# Underweight, overweight or obesity, diabetes, and hypertension in Bangladesh, 2004 to 2018

**Phuong Hong Nguyen**[1]*, **Salauddin Tauseef**[2], **Long Quynh Khuong**[3], **Rajat Das Gupta**[4,5], **Sk. Masum Billah**[6,7], **Purnima Menon**[1], **Samuel Scott**[1]

**1** International Food Policy Research Institute, Washington, DC, United States of America, **2** University of Manchester, Manchester, United Kingdom, **3** Hanoi University of Public Health, Hanoi, Viet Nam, **4** Arnold School of Public Health, University of South Carolina, Columbia, South Carolina, United States of America, **5** BRAC James P Grant School of Public Health, BRAC University, Dhaka, Bangladesh, **6** International Centre for Diarrheal Disease Research, Bangladesh (icddr, b), Dhaka, Bangladesh, **7** The University of Sydney School of Public Health, Sydney, New South Wales, Australia

* P.H.Nguyen@cgiar.org

## Abstract

### Background and objectives

Bangladesh is experiencing a nutrition transition with an increase in the double burden of malnutrition and non-communicable diseases (NCDs). This study sought to: 1) examine trends and differences in underweight, overweight/obesity, hypertension and diabetes by gender, area of residence, and wealth in Bangladesh from 2004 to 2018, 2) assess what factors contributed to changes in these outcomes.

### Methods

We used data from five rounds of the Bangladesh Demographic and Health Surveys (n = 76,758 women 15-49y and 10,900 men 18-95y in total). We calculated differences, slope index of inequality (SII) and concentration index (CIX) to examine trends over time and differences in outcomes by wealth and residence. We identified determinants and estimated drivers of changes in outcomes using regression-based decomposition.

### Results

Between 2004 and 2018, underweight prevalence decreased in both women (33% to 12%) and men (26% to 18%), whereas overweight/obesity increased (17% to 49% in women and 21% to 34% in men). Hypertension also increased in both women (31% to 44%) and men (19% to 33%) while diabetes changed marginally (11% to 14%). In all years, underweight was concentrated in poorer and rural households while overweight/obesity, diabetes and hypertension were concentrated in wealthier and urban households. Wealth inequity decreased over time for underweight, changed little for overweight/obesity, and increased for hypertension and diabetes among men. Increases in wealth explained 35% to 50% of the reduction in underweight and 30% to 57% of the increase in overweight/obesity.

**Editor:** Negar Rezaei, Non-Communicable Diseases Research Center, Endocrinology and Metabolism Population Sciences Institute, Tehran University of Medical Sciences, ISLAMIC REPUBLIC OF IRAN

**Data Availability Statement:** All relevant data are within the paper and its Supporting information files.

**Funding:** Funding for this research was provided by the Gates Foundation through Alive & Thrive, managed by FHI Solutions (grant numbers: OPP1170427). The funders had no role in study design, data collection and analysis, data interpretation, decision to publish, or preparation of the manuscript.

**Competing interests:** The authors have declared that no competing interests exist.

## Conclusion

Our findings imply that double duty actions are required to sustain the decrease in undernutrition and slow the increase in overweight/obesity and NCDs across diverse socioeconomic sections of the population in Bangladesh.

## Introduction

The Sustainable Development Goals (SDGs) aim to end all forms of malnutrition including undernutrition, overweight/ obesity, and diet-related noncommunicable diseases (NCDs) by 2030 [1]. Until the past decade, the focus has been on overweight/obesity in high income countries and undernutrition in low- and middle-income countries (LMICs) [2, 3]. While the prevalence of undernutrition remains high in many LMICs, especially in South Asia, trends in BMI suggest an emerging problem of overweight/obesity [2, 4]. Evidence from 126 LMICs with data from the 2010s suggests that 38% of these countries were facing a double burden of malnutrition (simultaneous undernutrition and overnutrition) based on a 20% overweight prevalence cutoff [3]. NCDs such hypertension and diabetes also disproportionately affect people living in LMICs [5].

Like many other LMICs, Bangladesh has been experiencing a nutrition transition in the past three decades alongside socio-economic development. Per capita income grew nearly three times between 1990s and 2010s, and the headcount measure of poverty fell from 49% in 2000 to 24% in 2016 [6]. Rising income and rapid urbanization were coupled with less physical activity, changing food systems and a shift from traditional diets to more unhealthy and processed foods [7, 8]. Overweight/obesity is also a leading risk factor of NCDs including cardiovascular disease, hypertension, and diabetes [9, 10]. The increase in the double burden of malnutrition and concurrent NCDs are major concerns for the overall health status of the country's population and for the health system [11]. The recent Global Burden of Diseases 2019 estimated that malnutrition, high blood pressure, high fasting plasma glucose and high body mass index are among the top 10 risk factors driving most death and disability in Bangladesh [12].

Recognizing the increase in overweight/obesity and NCDs as an emerging concern, some studies have examined trends in and risk factors of overweight/obesity or the double burden of underweight and overweight using multiple rounds of Bangladesh Demographic and Health surveys (BDHS) during 1996–2014 but have mainly focused on women [8, 13–17]. A few studies examined the prevalence of underweight, overweight/obesity in both women and men using a cross-sectional approach [18–20]. Studies using the 2017–2018 BDHS reported a double (underweight and overweight/obesity) and triple burden of NCDs (underweight, overweight/obesity and diabetes or hypertension) but did not link with previous survey rounds to capture trends and distributional patterns [9, 20–26]. Common factors associated with undernutrition or overnutrition in the literature include differences by residence (urban/rural) and wealth; however, there is no evidence on how these differences have changed over time. Understanding trends and patterns in malnutrition and NCDs is vital because it enables a better understanding of the geospatial and social distribution of these outcomes. Finally, there is limited information on which factors have contributed to the observed changes in undernutrition or overnutrition over time.

The present paper uses five rounds of BDHS data to: (1) examine changes in body mass index (BMI), underweight, overweight/obesity, diabetes, and hypertension among both women and men at national and subnational levels in Bangladesh from 2004 to 2018, (2)

quantify changes in differences in outcomes by wealth and residence (urban/rural) over time, and (3) understand what factors (such as improvements in wealth and education) contributed to changes in nutrition outcomes.

## Methods

### Data sources, sampling design and sample size

We used data from five rounds of BDHS (2004, 2007, 2011, 2014 and 2018) [27–31]. BDHS data are representative at both national and division levels. The BDHS surveys used a two-stage stratified sampling strategy, and a sampling frame was obtained from the list of enumeration areas of the Population and Housing Census of the People's Republic of Bangladesh, provided by the Bangladesh Bureau of Statistics [32]. In the first stage, enumeration areas were selected with probability proportional to size. A complete household listing operation was then carried out in all selected enumeration areas to provide a sampling frame for the second-stage selection of households. In the second stage, a systematic random sample of 30 households on average per enumeration area was selected. Details of the survey protocol and sampling are provided in BDHS reports [27–31].

Height and weight were measured for women in all five rounds, however the data for men were only available in two rounds (2011 and 2018). Blood pressure and blood glucose testing were measured in women and men aged 18 and older in a subsample of one-third of the households in 2011 and one-fourth of the households in 2018. Datasets are publicly available and can be downloaded from the DHS website (https://www.dhsprogram.com/) after obtaining permission from the DHS program.

We included all non-pregnant women or men with anthropometric or diabetes and hypertension data in the analysis. The sample size for this study is presented in Table 1. In total, 74,798 households were surveyed across the five rounds, among them 76,758 non-pregnant women aged 15-49y and 10,900 men aged 18-95y having height and weight measurement data. The number of non-pregnant women and men aged 35-95y having data on NCDs were 7,333 and 6,821, respectively.

### Variables

**Underweight and overweight/obese.** In the DHS surveys, anthropometric measurements for women and men were obtained by trained field staff using standard procedures [33, 34]. Weight was measured twice with light clothing and without shoes using a calibrated Seca 878 digital scale. Height was assessed three times without shoes using measuring boards made by Shorr Productions. The average of the three measurements was used in the analysis. BMI was calculated by weight (kg)/ height (m)$^2$. Underweight was defined as BMI $< 18.5$ kg/m$^2$ and overweight/obese as BMI $\geq 23$ kg/m$^2$, according to the criteria for Asian populations [35].

**Diabetes and hypertension.** Fasting blood glucose was tested with the HemoCue Glucose 201 analyzer (Teleflex Medical L.P., Markham, Canada) using a drop of capillary blood obtained from the middle or ring finger after respondents had fasted overnight. Fasting whole blood glucose measurements were then converted to fasting plasma glucose equivalent values [36]. Respondents were classified as diabetic if they had fasting plasma glucose level $\geq 7$ mmol/L as recommended by the World Health Organization (WHO) [37] or if they reported currently taking prescribed medication for high blood glucose or diabetes.

Systolic blood pressure and diastolic blood pressure were measured with the Life Source UA-767 Plus digital blood pressure monitor (A&D Medical, San Jose, USA). Blood pressure was measured three times, with an interval of approximately 10 minutes between measurements. The average value of the second and third measurements was used for our analyses.

**Table 1. Characteristics of households, women and men by survey round, Bangladesh 2004–2018.**

| | 2004 | 2007 | 2011 | 2014 | 2018 |
|---|---|---|---|---|---|
| **Sample size, *n* (%$^{\$}$)** | | | | | |
| Number of households | 10,500 (14.1) | 10,400 (13.9) | 17,141 (22.9) | 17,300 (23.1) | 19,457 (26.0) |
| Number of women 15-49y | 14,921 (14.8) | 14,578 (14.4) | 23,038 (22.8) | 22,812 (22.6) | 25,757 (25.4) |
| Number of non-pregnant women 15-49y with BMI data | 11,459 (14.9) | 11,118 (14.5) | 16,663 (21.7) | 17,964 (23.4) | 19,554 (25.5) |
| Number of non-pregnant women 35-95y with diabetes and hypertension data | -- | -- | 3,753 (51.2) | -- | 3,580 (48.8) |
| Number of men 18-95y | 14,196 (15.5) | 13,993 (15.2) | 21,012 (22.9) | 20,490 (22.3) | 22,071 (24.1) |
| Number of men 18-95y with BMI data | -- | -- | 5,247 (51.9) | -- | 5,653 (48.1) |
| Number of men 35-95y with diabetes and hypertension data | -- | -- | 3,692 (54.1) | -- | 3,129 (45.9) |
| **Characteristics of sample for BMI analyses** | | | | | |
| **Household** | | | | | |
| Place of residence (rural), % | 77.7 | 77.6 | 75.0 | 72.0 | 72.3 |
| Household size, *n* (mean ± SD) | 5.6±2.8 | 5.5±2.7 | 5.2±2.5 | 5.1±2.4 | 5.1±2.5 |
| SES index (0–10), score (mean ± SD) | 2.1±2.3 | 4.1±2.7 | 5.2±2.5 | 5.7±2.4 | 6.1±2.1 |
| Exposed to media, % | 4.9 | 46.7 | 47.2 | 48.1 | 50.2 |
| Improved latrine, % | 60.5 | 41.8 | 54.8 | 70.2 | 64.2 |
| Improved drinking water, % | 95.6 | 97.3 | 98.5 | 97.6 | 98.2 |
| **Women 15-49y** | | | | | |
| Age, y (mean ± SD) | 30.0±9.3 | 30.4±9.3 | 31.2±9.2 | 30.8±9.1 | 31.6±9.2 |
| Education, y (mean ± SD) | 3.4±3.7 | 4.0±3.8 | 4.7±3.9 | 5.1±4.0 | 5.6±4.0 |
| Currently working, % | 21.5 | 32.7 | 14.2 | 33.9 | 49.1 |
| Currently married, % | 100 | 100 | 100 | 100 | 98.2 |
| Having children <5y, % | 58.1 | 54.7 | 51.3 | 49.2 | 47.8 |
| **Men 18-95y** | | | | | |
| Age, y (mean ± SD) | -- | -- | 45.2±15.3 | -- | 41.6±16.7 |
| Education, y (mean ± SD) | -- | -- | 4.6±4.5 | -- | 5.6±4.7 |
| Manual worker, % | -- | -- | 50.9 | -- | 53.0 |
| Currently married | -- | -- | 99.7 | -- | 83.8 |
| **Characteristics of sample for diabetes and hypertension analyses** | | | | | |
| **Household** | | | | | |
| Place of residence (rural), % | -- | -- | 75.8 | -- | 74.0 |
| Household size, *n* (mean ± SD) | -- | -- | 5.1±2.4 | -- | 4.9±2.3 |
| SES index (0–10), score (mean ± SD) | -- | -- | 5.2±2.6 | -- | 6.0±2.2 |
| Exposed to media, % | -- | -- | 47.2 | -- | 49.2 |
| Improved latrine, % | -- | -- | 55.4 | -- | 64.7 |
| Improved drinking water, % | -- | -- | 98.4 | -- | 98.0 |
| **Women 35-95y** | | | | | |
| Age, y (mean ± SD) | -- | -- | 45.8±10.3 | -- | 45.1±9.5 |
| Education, y (mean ± SD) | -- | -- | 2.7±3.7 | -- | 3.3±3.9 |
| Currently working, % | -- | -- | 12.9 | -- | 52.3 |
| Currently married, % | -- | -- | 99.9 | -- | 99.9 |
| Having children <5y, % | -- | -- | 31.6 | -- | 29.7 |
| **Men 35-95y** | | | | | |
| Age, y (mean ± SD) | -- | -- | 51.9±12.8 | -- | 51.5±13.2 |
| Education, y (mean ± SD) | -- | -- | 4.4±4.7 | -- | 4.6±4.7 |
| Manual worker, % | -- | -- | 50.9 | -- | 55.4 |
| Currently married, % | -- | -- | 99.6 | -- | 99.5 |

$^{--}$ Data not available;

$^{\$}$Proportion of sample from specific round to the total sample size;

BMI: Body mass index; SES: Socio-economic status

Respondents were considered as hypertensive if they had systolic blood pressure level $\geq 140$ mmHg or diastolic blood pressure level $\geq 90$ mmHg, or if they were currently taking antihypertensive medication [38].

**Determinants of malnutrition, diabetes and hypertension.** We selected all relevant known determinants of undernutrition, overnutrition, diabetes and hypertension that were available in BDHS [39–41]. Individual level variables included age, education, working status, marital status, having children <5 years (for women) and exposure to mass media (defined as if the respondent reads the newspaper, listens to the radio, or watches TV at least once a week). Household level variables included household size, place of residence (rural/urban), improved latrine, access to improved drinking water, and wealth. A wealth index was constructed on pooled data from 2004 to 2018 using a principal component extracted from multiple variables including household ownership of different assets, livestock, house and land, and key housing characteristics [42]. The first component derived from the component scores was then used to categorize wealth into quintiles.

## Data analysis

Descriptive statistics were used to summarize sample characteristics, with percentages for categorical variables, and mean and standard deviation for continuous variables. All estimations were adjusted for the survey-specific sampling design using the *"svy"* Stata package.

Data were pooled from different survey rounds to estimate trends in both mean BMI and prevalence of overweight/obesity, diabetes and hypertension from 2004 to 2018. We used K-density plots to visualize the distribution of BMI, and locally weighted smoothing regression to examine differences in BMI by respondent age. We plotted prevalence of overweight/obesity, diabetes and hypertension by survey year and used maps to illustrate sub-national variability in the prevalence of outcomes for each survey rounds. We also calculated average annual rate change (AARC) based on a formula adapted from UNICEF [43] (S1 Table).

To examine differences in the prevalence of underweight, overweight/obesity, diabetes, and hypertension by population subgroup over time, we first used equity plots disaggregated by wealth quintile and residential area (rural/urban). We then examined absolute and relative differences for each health outcome by wealth quintile using the slope index of inequality (SII) and the concentration index (CIX). These two complex measures consider the entire distribution of outcomes over the five wealth quintiles and are weighted by the sample size of each quintile [44, 45]. Both SII and CIX range from -100 to +100, with positive values indicating that the outcome is more prevalent among the rich and negative values indicating that the outcome is more prevalent among the poor.

To estimate the contributions of changes in determinants to changes in health outcomes over time, we first used multivariable logistic regressions to examine associations between determinants and outcomes. We then applied a regression decomposition analysis to assess how changes in key determinants can predict changes in underweight and overweight/obesity from 2004 to 2018 for women, and from 2011 to 2018 for men. This regression decomposition combines the analysis of differences in means of the determinant variables over time and regression estimates of the coefficients associated with these variables from the regression model. For example, if a determinant has a large regression coefficient associated with outcome and a large change over time, then this determinant will play a large role in explaining changes in outcome over time. We did not conduct decomposition analysis for diabetes given the small changes in this outcome. All models controlled for the survey year, division fixed effects, cluster sampling design and sampling weights used in the survey. All analyses were conducted using Stata 17 (StataCorp LLC, College Station, TX, USA). Statistical significance was considered at $p < 0.05$.

### Ethics statement

This study was a secondary data analysis of BDHS data, which was approved by the institutional review board at ICF and the Bangladesh Medical Research Council. All respondents in the BDHS undergo an informed consent process for participation in the survey. Specifically, the enumerators read a detailed informed consent statement to the respondents, informing about the survey, describing the procedure, and emphasizing the voluntary nature of participation. Respondents signed the consent form once they agreed to participate in the survey.

## Results

### Sample characteristics

On average across survey rounds, women were 30 years old, men were 42–45 years old and most (72–78%) lived in rural areas (Table 1). Over time, household size reduced from 5.6 to 5.1 members and wealth index increased from 2.1 to 6.1. Hygiene and sanitation also improved by 22 percentage points (pp). Average years of education increased for women (from 3.4 years in 2004 to 5.6 years in 2018) and men (4.6 years in 2011 to 5.6 years in 2018). Women participating in labor workforce also increased over time.

### Trends in BMI, underweight, overweight/obesity, hypertension and diabetes

Mean BMI in women increased from 20.2 kg/m$^2$ in 2004 to 23.1 kg/m$^2$ in 2018, and in men from 20.5 kg/m$^2$ in 2011 to 21.6 kg/m$^2$ in 2018 (Fig 1A). The increase was faster in women (0.26 kg/m$^2$ per year) than men (0.16 kg/m$^2$ per year) between 2011 and 2018. Mean BMI increased with age in both women and men until the age of 35 years (Fig 1B). After 35 years, the BMI curve exhibited a slight downward trend in women and a stronger downward trend in men. Correspondingly, overall underweight gradually reduced among women and men under aged 35 years and increased when they get older (S1 Fig). Opposite trends were observed for overweight/obesity.

While the prevalence of underweight among women steadily declined (from 33% in 2004 to 12% in 2018), the prevalence of overweight/obesity rose nearly three times (from 17% in 2004 to 49% in 2018) (Fig 1C). In 2018, overnutrition was four times as prevalent as undernutrition in women. A similar pattern of declining underweight and increasing overweight prevalence was observed for men, with 20% of men being underweight and 33% of men being overweight in 2018. The AARC in underweight was -6.8% for women and -4.9% for men, and the AARC in overweight/obesity was +7.8% for women and +7.4% for men (S1 Table).

Hypertension increased over time and with increasing age in both women and men (Fig 2A). The prevalence of hypertension increased by 12.7 pp and 14.6 pp from 2011 to 2018 among women and men, respectively, reaching 44% in women and 33% in men in 2018 (Fig 2C). Diabetes prevalence exhibited a slow upward trend, increasing from 11% to 14% in both men and women (Fig 2B). The AARC for hypertension was +5.0% for women and +8.6% for men, while the AARC for diabetes was +3.2% for women and +4.0% for men.

Changes in underweight and overweight/obesity varied by division of the country (Fig 3A). Among women, between 2004 and 2018, the largest reductions in underweight occurred in Barisal (-25 pp) and Chittagong (-24 pp) while the largest increases in overweight/obesity occurred in Chittagong (+38 pp), Barisal and Dhaka (each +34 pp). Similar findings were observed for men between 2011 and 2018 (-14 pp for underweight in Barisal, +17 to +18 pp for overweight/obesity in Barisal, Chittagong and Dhaka).

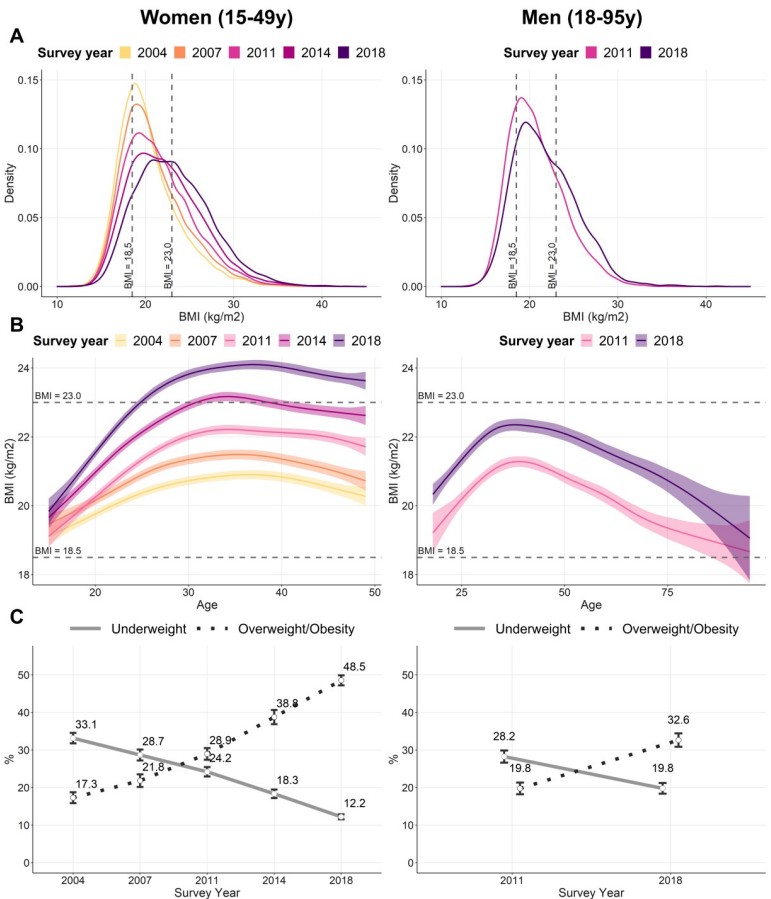

**Fig 1. Trends in BMI, overweight/obesity, and underweight among women and men by survey round, Bangladesh 2004–2018[1].** (A) Distributions of BMI by survey round. (B) Distributions of BMI by age and survey round. (C) Prevalence of underweight and overweight/obesity by survey round. [1]Underweight was defined as BMI < 18.5 kg/m$^2$ and overweight/obese as BMI ≥ 23 kg/m$^2$, according to the criteria for Asian populations. Values in (A) and (B) are mean and 95% confidence interval bands. Values in (C) are mean and 95% confidence interval points.

There was less divisional variation in change in diabetes, but Dhaka showed the largest increase in diabetes for both women (+8 pp) and men (+11 pp) (Fig 4). Hypertension showed largest increase in Barisal, and Sylhet Chittagong among women and men (+17 to +22 pp).

## Wealth and residential differences in underweight, overweight/obesity, hypertension and diabetes

Underweight was higher among the poor compared to the rich across gender and survey rounds (Fig 5 and S2 Table). The wealth gap in underweight was large for both men and women although the gap reduced in recent years; SII decreased from -36 pp in 2004 to -19 pp in 2018 for women, and from -37pp in 2011 to -26 pp in 2018 for men. The opposite story emerged for overweight/obesity which was higher among wealthier households. The wealth gap in overweight/obesity was large (SII: 44 to 56 pp for women and 50–52 pp for men) and changed minimally over time.

Hypertension and diabetes were also more prevalent in richer households. In women, the wealth gap in hypertension decreased (SII decreased from 22 to 14 pp), but the gap in diabetes

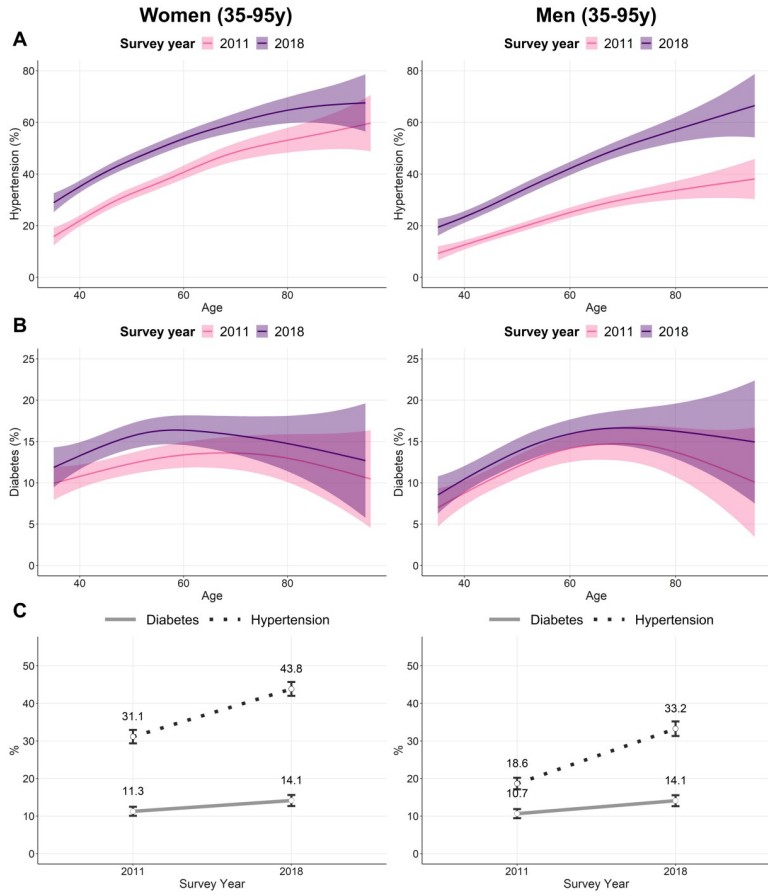

**Fig 2. Trends in hypertension and diabetes among women and men by survey rounds, Bangladesh 2004–2018.** (A) Prevalence of hypertension by age and survey round. (B) Prevalence of diabetes by age and survey round. (C) Trends in hypertension and diabetes. Values in (A) and (B) are mean and 95% confidence interval bands. Values in (C) are mean and 95% confidence interval points.

increased (SII increased from 18 to 23 pp). In men, wealth gaps increased for both hypertension and diabetes (SII increased from 23 to 25 pp and 16 to 25 pp, respectively).

With respect to place of residence (Fig 5 and S3 Table), the prevalence of underweight was higher but overweight, hypertension and diabetes were lower in rural compared to urban areas. The rural-urban gaps reduced for all indicators over time, with a more prominent reduction in underweight compared to overweight.

## Associations between selected determinants and health outcomes

Findings from bivariate analysis (S4 Table) and multivariable regression analysis (Table 2) using data pooled from all available survey rounds confirmed the association between household SES and outcomes for both women and men. Compared to those in poorest quintile, women in the highest quintile were 70% less likely to be underweight (adjusted odds ratio 0.30, 95% confidence interval 0.26–0.33), ~4 times more likely to be overweight, 2.6 times more likely to have diabetes, and 1.4 times more likely to have hypertension. Similar associations were observed for men. Rural residence predicted underweight and overweight among women but not men. Women and men with higher education had lower odds of being

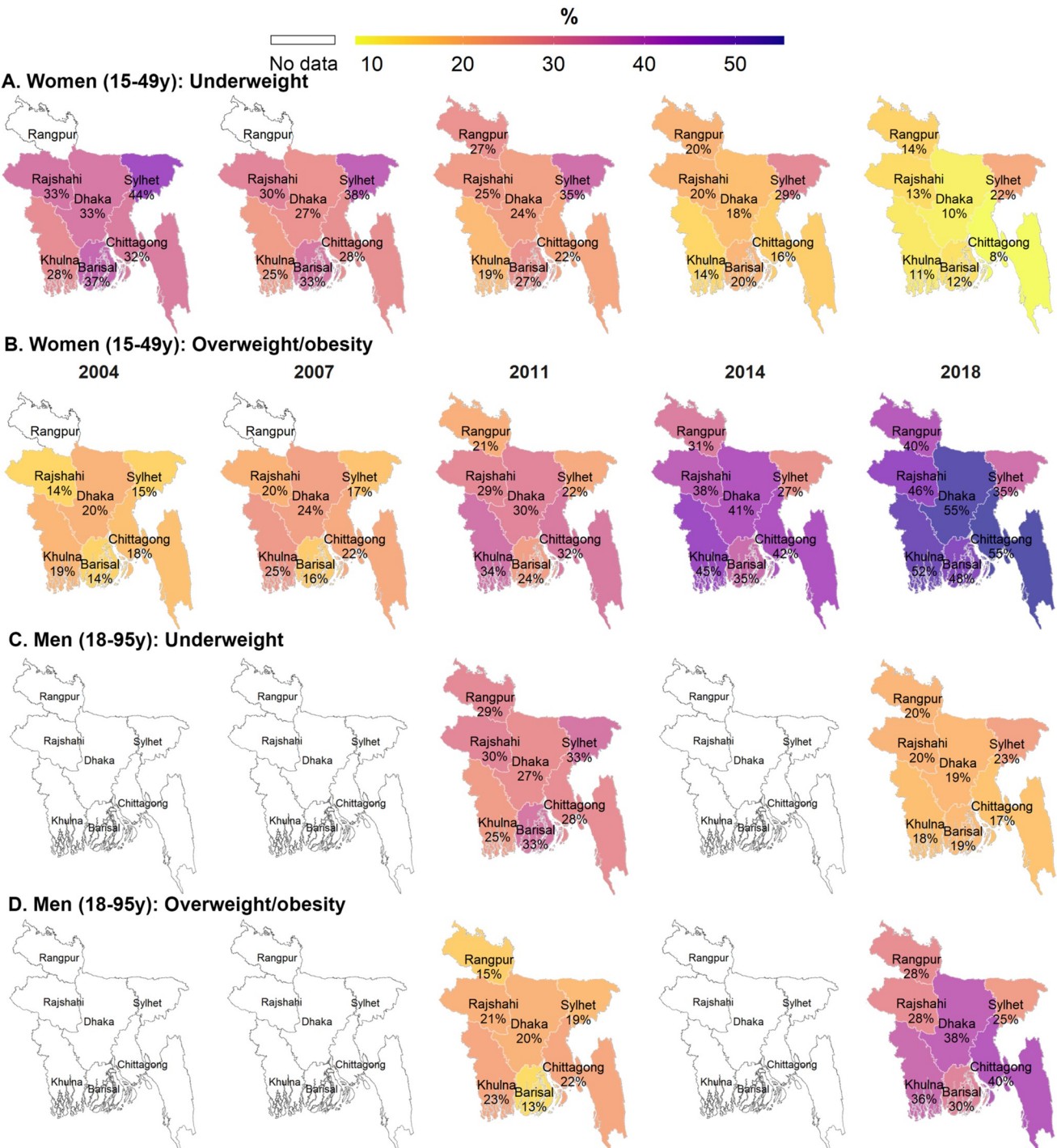

**Fig 3. Prevalance of underweight and overweight/obesity among women and men by division and survey round, Bangladesh 2004–2018.** (A) Women (15-49y): Underweight. (B) Women (15-49y): Overweight/obesity. (C) Men (18-95y): Underweight. (D) Men (18-95y): Overweight/obesity. Note: This map is created from the shapefile (an empty map) that is publicly available from http://gadm.org.

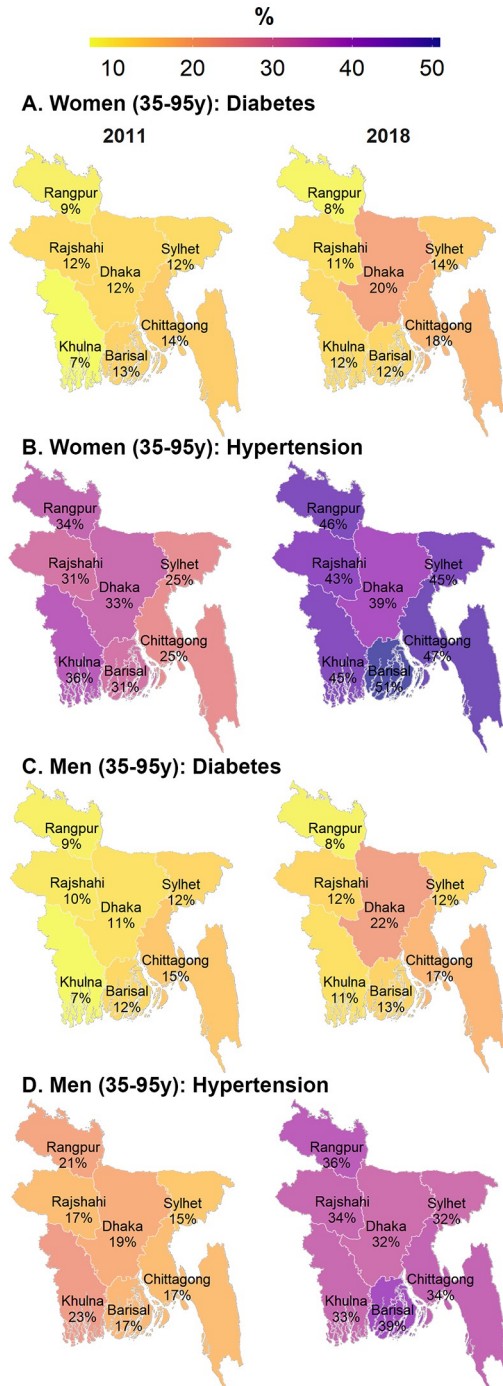

**Fig 4. Prevalence of noncommunicable diseases among women and men by division and survey round, Bangladesh 2004–2018.** (A) Women (35-95y): Diabetes. (B) Women (35-95y): Hypertension. (C) Men (35-95y): Diabetes. (D) Men (35-95y): Hypertension. Note: This map is created from the shapefile (an empty map) that is publicly available from http://gadm.org.

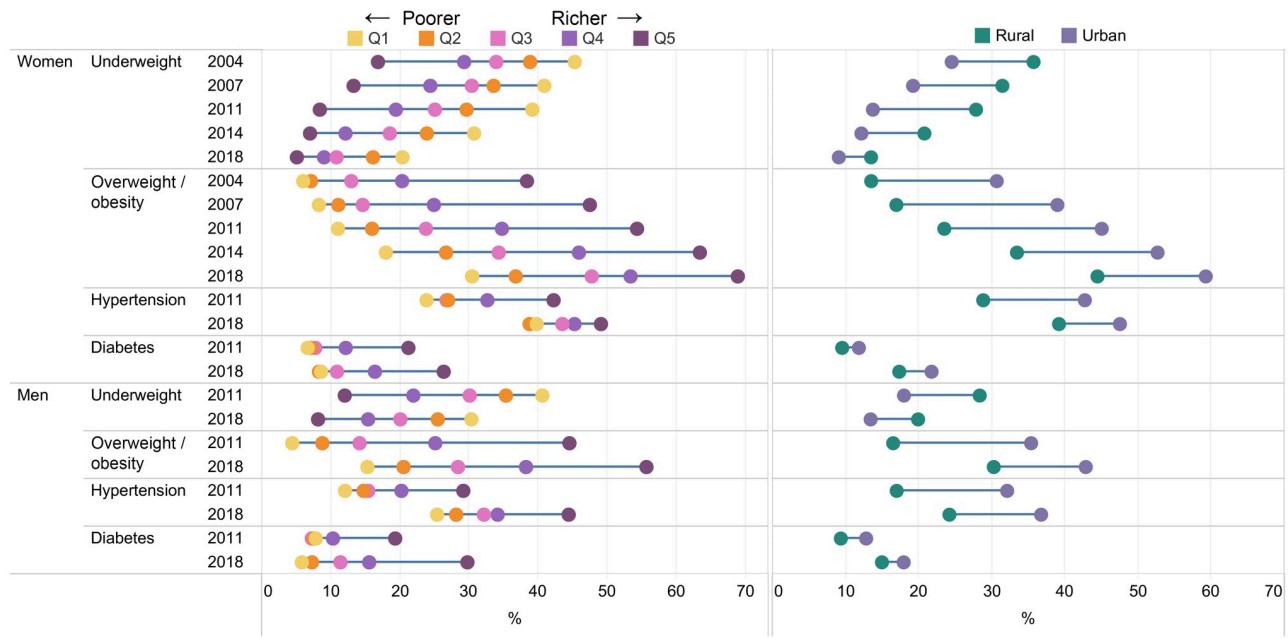

**Fig 5. Socio-economic and residential inequality in underweight, overweight/obesity, diabetes and hypertension among women and men by survey round, Bangladesh 2004–2018.**

underweight and higher odds of overweight/obesity, diabetes and hypertension (except for hypertension among women).

## Regression decomposition to identify drivers of changes in underweight and overweight/obesity

Findings from the regression models in Table 2 were then combined with the changes over time in determinants to estimate the extent to which changes in determinants contributed to changes in outcomes over time. The decomposition models explained 60% of the actual change in underweight and 73% change in overweight/obesity in women (Fig 6). The explained share in the reduction of underweight and increase in overweight for women was mostly accounted for by the improvement in wealth (50% and 57% of changes respectively), followed by the improvement in education (5% and 8% respectively). For men, the decomposition models explained 56% changes in underweight and 45% changes in overweight/obesity, with the two key contributions for the changes in underweight and overweight/obesity being improvement in wealth (35% and 30%, respectively) and education (10–12%). Decomposition analysis only explained 40% and 20% of the increase in hypertension in women and men, respectively, where the increase in overweight/obesity explained 30% of the increase in hypertension among women and 16% among men.

## Discussion

Using data from multiple nationally representative health surveys spanning 14 years, we examined trends in underweight, overweight/obesity, diabetes and hypertension among both women and men between 2004 and 2018, and we identified some factors that have contributed to changes in outcomes over time. We found that prevalence of underweight in both women and men declined substantially between 2004 and 2018 but remains higher in the poorer and rural compared to richer and urban households. In contrast, the prevalence of overweight/

**Table 2. Associations between selected factors and outcomes.**

| | Outcomes | | | |
|---|---|---|---|---|
| | Underweight | Overweight/obesity | Diabetes | Hypertension |
| Predictors | OR (95% CI) | OR (95% CI) | OR (95% CI) | OR (95% CI) |
| **Women** | | | | |
| Household level | | | | |
| Household size | 1.01* (1.00,1.02) | 0.98*** (0.97,0.99) | 0.98 (0.95,1.02) | 0.97* (0.95,1.00) |
| Household SES (ref: Quintile 1) | | | | |
| Quintile 2 | 0.79*** (0.74,0.84) | 1.22*** (1.13,1.32) | 0.97 (0.69,1.35) | 1.08 (0.90,1.30) |
| Quintile 3 | 0.67*** (0.63,0.72) | 1.58*** (1.45,1.71) | 1.14 (0.82,1.59) | 1.1 (0.89,1.35) |
| Quintile 4 | 0.54*** (0.50,0.60) | 2.15*** (1.95,2.37) | 1.67** (1.17,2.38) | 1.15 (0.91,1.46) |
| Quintile 5 | 0.30*** (0.26,0.33) | 3.94*** (3.54,4.39) | 2.63*** (1.78,3.86) | 1.42* (1.07,1.87) |
| Exposed to media | 0.86*** (0.81,0.91) | 1.24*** (1.18,1.31) | 1.01 (0.80,1.27) | 1.06 (0.91,1.24) |
| Rural | 1.12*** (1.05,1.20) | 0.80*** (0.75,0.85) | 0.88 (0.73,1.07) | 0.89 (0.77,1.03) |
| Hygiene and sanitation | | | | |
| Improved latrine | 0.89*** (0.85,0.94) | 1.15*** (1.10,1.21) | 0.87 (0.72,1.05) | 0.99 (0.87,1.12) |
| Improved drinking water | 0.92 (0.80,1.05) | 1.26** (1.07,1.47) | 1.15 (0.69,1.91) | 1.07 (0.70,1.63) |
| Individual level | | | | |
| Age | 0.98*** (0.98,0.99) | 1.05*** (1.05,1.05) | 1.01*** (1.01,1.02) | 1.05*** (1.04,1.05) |
| Education (ref: no education) | | | | |
| ≤ Primary school | 0.82*** (0.77,0.87) | 1.44*** (1.36,1.53) | 1.2 (0.98,1.49) | 0.92 (0.80,1.06) |
| ≤ Secondary school | 0.72*** (0.67,0.77) | 1.83*** (1.71,1.96) | 1.30* (1.00,1.69) | 1.03 (0.86,1.24) |
| College or higher | 0.60*** (0.53,0.67) | 2.07*** (1.90,2.27) | 1.06 (0.73,1.55) | 0.68* (0.50,0.92) |
| Currently working | 1.04 (0.98,1.09) | 0.85*** (0.81,0.90) | 0.79* (0.64,0.98) | 0.86* (0.75,0.99) |
| Having children <5y | 1.14*** (1.09,1.19) | 0.88*** (0.84,0.92) | 0.89 (0.73,1.08) | 0.87* (0.76,0.99) |
| Overweight/obesity | - - - - | - - - - | 1.75*** (1.46,2.09) | 2.60*** (2.29,2.96) |
| **Men** | | | | |
| Household level | | | | |
| Household size | 0.99 (0.97,1.01) | 1.02 (0.99,1.04) | 0.98 (0.95,1.01) | 0.97* (0.95,1.00) |
| Household SES (ref: Quintile 1) | | | | |
| Quintile 2 | 0.82* (0.70,0.97) | 1.30* (1.03,1.64) | 1.04 (0.74,1.48) | 1.13 (0.90,1.41) |
| Quintile 3 | 0.69*** (0.58,0.83) | 1.64*** (1.30,2.08) | 1.21 (0.83,1.75) | 1.14 (0.89,1.47) |
| Quintile 4 | 0.51*** (0.41,0.65) | 2.46*** (1.91,3.16) | 1.60* (1.08,2.37) | 1.22 (0.92,1.61) |
| Quintile 5 | 0.28*** (0.21,0.39) | 4.20*** (3.18,5.56) | 3.25*** (2.11,5.00) | 1.65** (1.20,2.26) |
| Expose to media | 0.93 (0.80,1.09) | 1.06 (0.91,1.23) | 1.03 (0.81,1.29) | 0.94 (0.79,1.12) |
| Rural | 1.09 (0.94,1.28) | 0.94 (0.81,1.08) | 1.30* (1.06,1.60) | 0.95 (0.80,1.13) |
| Hygiene and sanitation | | | | |
| Improved latrine | 0.94 (0.82,1.07) | 1.26*** (1.10,1.45) | 0.88 (0.71,1.09) | 1 (0.86,1.17) |
| Improved drinking water | 1.39 (0.90,2.16) | 0.81 (0.62,1.08) | 1.27 (0.68,2.37) | 0.76 (0.54,1.08) |
| Individual level | | | | |
| Age | 1.01*** (1.01,1.02) | 1 (1.00,1.01) | 1.02*** (1.01,1.03) | 1.04*** (1.04,1.05) |
| Education | | | | |
| ≤ Primary school | 0.81** (0.70,0.94) | 1.51*** (1.27,1.80) | 1.33* (1.05,1.67) | 1.19* (1.00,1.40) |
| ≤ Secondary school | 0.70*** (0.59,0.83) | 1.92*** (1.59,2.31) | 1.15 (0.88,1.49) | 1.40*** (1.15,1.69) |
| College or higher | 0.45*** (0.35,0.58) | 2.91*** (2.34,3.63) | 1.50* (1.10,2.03) | 1.73*** (1.36,2.19) |
| Manual worker | 1.04 (0.91,1.19) | 0.68*** (0.61,0.77) | 0.84 (0.69,1.01) | 0.74*** (0.64,0.85) |

*(Continued)*

**Table 2.** (Continued)

| Predictors | Outcomes | | | |
| --- | --- | --- | --- | --- |
| | Underweight | Overweight/obesity | Diabetes | Hypertension |
| | OR (95% CI) | OR (95% CI) | OR (95% CI) | OR (95% CI) |
| Overweight/obesity | - - - - | - - - - | 1.69*** (1.40,2.05) | 2.28*** (1.95,2.66) |

\* $p < 0.05$,

\*\* $p < 0.01$,

\*\*\* $p < 0.001$

obesity increased nearly three-fold among women and 1.5 times among men, affecting nearly half of women and a third of men in 2018. Hypertension also increased (by 13–15 pp) while diabetes changed marginally over time. Changes in outcomes also varied by subnational division. When examining differences in outcomes across the wealth spectrum, wealth gaps decreased over time for underweight, remained stable for overweight/obesity, and increased for diabetes and hypertension s among men. Increasing wealth was the key driver of change of the reduction in undernutrition and of increase in overweight/obesity.

Our findings of a decreasing trend in the prevalence of underweight and an increasing trend of overweight/obesity are well aligned with studies using previous rounds of BDHS data [8, 13–17, 46, 47]. Studies examining changes in BMI [13], overweight/obesity [15, 16] or the double burden of under- and overnutrition [8, 14, 17] mainly focus on women of reproductive age, showing that the rates of change varying across residential areas and wealth quintile stratum. Our study extends these analyses to the most recent BDHS round in 2017–2018 and

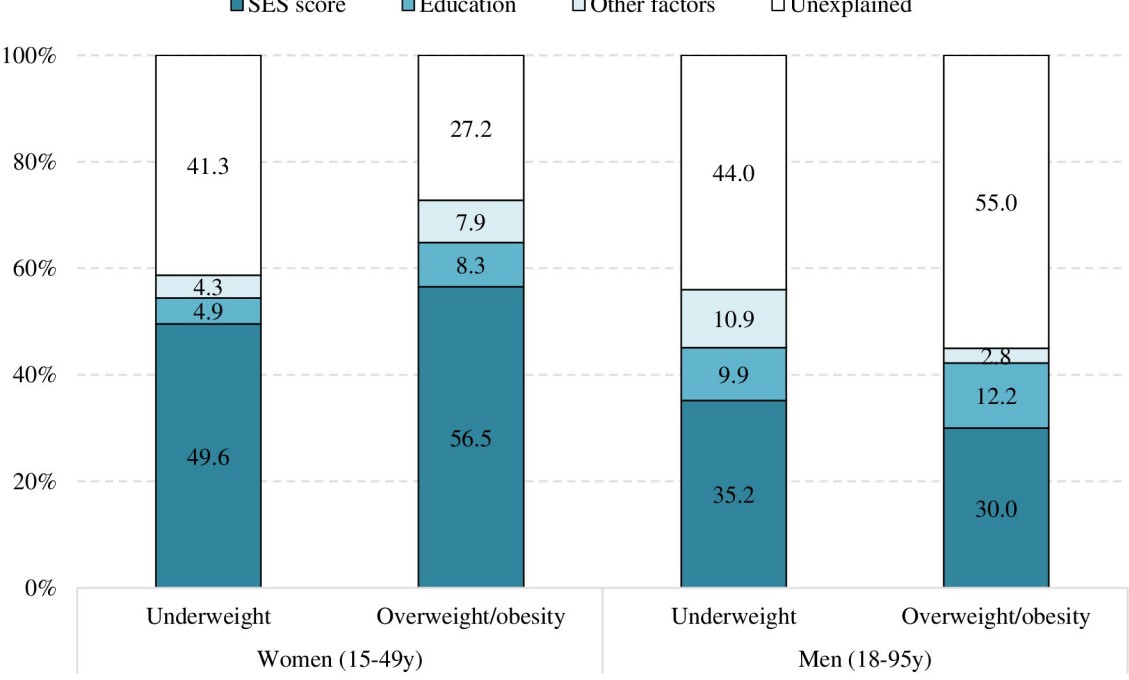

**Fig 6. Regression decomposition analysis of factors that explain changes in underweight and overweight/obesity in women (2004–2018) and men (2011–2018).** Other factors include hygiene and sanitation in men, and household size, hygiene and sanitation, age, working status, and having children under 5 year in women.

examines changes in malnutrition and diabetes and hypertension in both men and women. We found a higher AARC for underweight among women than men (-7% vs. -5%), similar AARC for overweight/obese (+7%) and lower AARC for hypertension (+5% vs. +9%) and diabetes (+3% vs. +4%).

The changes in nutrition and health outcomes are congruent with rapid economic growth and poverty reduction in Bangladesh over the past two decades [48]. As observed in other similar developing LMICs [49], Bangladesh has been experiencing transitions in the food supply, household food expenditures, diets, and nutrition outcomes. Food supply per person increased between 2010 and 2019 for most food groups, including eggs (84%), vegetables (46%), vegetable oils (42%), pulses (36%), among others (S5 Table) (FAO Statistical Database on Food Balance, http://www.fao.org/faostat/en/#data/FBS) [50]. Between 2010 and 2016, the monthly average household consumption increased 39% (from Tk. 11,000 to Tk. 15,240) [51]. While overall calorie intake per capita per day has remained approximately the same (2240 Kcal in 1996 and 2210 Kcal in 2016), there have been large increases in daily per capita consumption of animal source foods (chicken 4g to 17g; eggs 3g to 14g) and edible oil (10g to 27g) [51]. There has also been a rapid emergence of the fast-food industry, especially in urban areas [52–54]. Dietary changes have been accompanied by lifestyle changes. Nationally representative and internationally comparable data from 2010 (using Global Physical Activity Questionnaire Version 2) on physical activities in Bangladesh showed that more than one-third of adults had a low physical activity level; inactivity was higher in females than males (54% vs. 15%), and higher in urban compared to rural areas (38% vs. 32%) [55].

Improvements in wealth and its associated changes have contributed to reductions in underweight in Bangladesh [8, 14, 56] and in other similar study contexts such as India [49]. However, improvements in wealth can be a "double edged sword" in the fight for nutrition and against NCDs since such improvements have been associated with increases in overweight, hypertension, and diabetes [49], as seen in the present study. In contrast to developed nations, overweight/obesity is more prevalent in higher-income groups than in lower-income groups in South Asia [57]. This is congruous with a global study on 36 LMICs which showed a consistent positive association between wealth index and BMI [58]. Higher-income households have increased purchasing power and access to food, and are less likely to face intra-household food allocation disparity [59]. However, the diet consists predominantly of carbohydrate-rich staples such as white rice [51, 60], the excess consumption of which has been shown to be associated with overweight/obesity, type 2 diabetes and metabolic syndrome [61, 62]. The 2016 Household Income and Expenditure Survey in Bangladesh revealed that poverty fell substantially between 2000 and 2016, shown as an increase in household monthly income, expenditure, and food consumption [6]. However, per capita per day intake of major food items (such as pulses, vegetables, fruits, meat, fish, eggs and milk) among poor households was much lower than in non-poor households; this pattern also held for edible oils (18.8 vs. 29.3 grams in non-poor and poor households, respectively) and sugar (2.7 vs. 8.2 grams) [6]. Higher income groups are also less likely to engage in physically demanding work [63].

Acknowledging the alarming increase in overweight/obesity and hypertension along with the steady reduction in underweight, the Government of Bangladesh seeks to address malnutrition in all forms per the National Plan of Action for Nutrition 2016–2025 [64]. The perils of the double burden of malnutrition have been acknowledged in the plan, with specified action areas, activities, and indicators proposed to monitor progress. A policy analysis in 2017 identified 51 policy documents in Bangladesh, which is aligned with the World Health Organization's 2013–2020 Action Plan for the Global Strategy for the Prevention and Control of NCDs [65]. There are at least 8 national policies, 12 laws, and 3 strategic plans which address the reduction of modifiable risk factors and promotion of a healthy environment [65]. Recently,

the Government of Bangladesh has adopted a 'Multisectoral Action Plan for Prevention and Control of Noncommunicable Diseases 2018–2025' [66], which aims to reduce overweight and obesity through the promotion of physical activity and a healthy diet. However, the implementation of these policies and plans remains slow in progress. To our knowledge, there is no nationwide, multisectoral package of interventions to address the rise of overweight/obesity.

The present study has several strengths. It examines trends in underweight and overweight/obesity as well as diabetes and hypertension, using a large sample of adult women and men from five rounds of nationally representative survey spanning 14 years. Using advanced statistical methods, we describe changes and identify determinants of change over time across rural-urban residence and wealth quintiles disaggregated by gender. However, some limitations in the analysis warrant discussion. The identification of the key drivers of change of underweight, overweight/obesity, diabetes and hypertension are limited by the availability of variables in the DHS datasets. For example, more than 40% of the change in underweight for both genders and in overweight for men could not be explained by the variables considered in the regression analysis. Information on physical activities of individuals and dietary intake could have been vital in identifying the "unexplained" factors in this analysis. Decomposition analysis of drivers of change was also not possible for diabetes given that diabetes has increased by only 3–4 pp over time.

## Conclusion

While Bangladesh observed remarkable progress on reducing underweight, differences across geography, residential and socioeconomic groups remain, and the increases in overweight/obesity, diabetes and hypertension are concerning. As a country with a strong record of growth and poverty reduction, Bangladesh is expected to continue to see increasing wealth, development, and urbanization. While these changes can bring benefits to living conditions, they are likely to continue to bring challenges such as increasing overweight/obesity, diabetes and hypertension. These changes could burden health systems and contribute to the overall burden of disease for the nation and for individuals. We also note the substantial lack of data on diets and physical activity, both critical immediate determinants of overweight and NCDs, and call for inclusion of these data in large-scale national surveys to strengthen such analyses. Double duty actions with multidisciplinary targeted interventions addressing the broad set of underlying factors should be explored to sustain the decrease in undernutrition and slow the increase in overweight/obesity and NCDs in Bangladesh.

## Supporting information

**S1 Fig. Prevalance of underweight and overweight/obesity among women and men by age group, Bangladesh 2004–2018.**
(DOCX)

**S1 Table. Estimating average annual rate of change (AARC) in underweight, overweight/ obesity, diabetes and hypertension in women (2004–2018) and men (2011–2018).** The following methods are adapted from UNICEF (42). When prevalence estimates are available for multiple years in a country, the AARC can be calculated using a regression analysis as follows: If the prevalence in a baseline year t0 is Y0 and four data points after t0 are available for trend analysis, then each of the four points can be written as: $Yti = Y0^{*}(1-b\%)^{(ti-t0)}$, so that, $\ln(Yti) = \ln(Y0) + (ti-t0)^{*}\ln(1-b\%) = \ln(Y0) + ti^{*}\ln(1-b\%) - t0^{*}\ln(1-b\%) = \beta^{*}ti + C0$, Where $\beta = \ln(1-b\%)$ and $C0 = \ln(Y0) - t0^{*}\ln(1+b\%)$, a constant $\beta$, the coefficient of ti, in a simple linear

regression of ln(Yi) against ti can then be translated into b%. The AARC = (EXP(β) *100)-100.
(DOCX)

**S2 Table. Socio-economic inequality in underweight, overweight/obesity, diabetes and hypertension among women and men by survey round, Bangladesh 2004–2018.** [1]Negative values mean that the burden is more concentrated in the poor and positive values mean that the burden is more concentrated in the wealthy. *,**,*** Significant difference for inequality between Q1 and Q5: *P<0.05*, **P<0.01, *** *P* < 0.001; Q: quintile. SII: Slope Index of Inequality, CIX: Concentration Index.
(DOCX)

**S3 Table. Residential inequality in underweight, overweight/obesity, diabetes and hypertension among women and men by survey round, Bangladesh 2004–2018.** [1]Negative values mean that the burden is more concentrated in the rural and positive values mean that the burden is more concentrated in the urban. *,**,*** Significant difference for inequality between rural and urban areas: *P<0.05*, **P<0.01, *** *P* < 0.001; SII: Slope Index of Inequality, CIX: Concentration Index.
(DOCX)

**S4 Table. Number and percentage of individuals with the outcomes, by selected factors.**
(DOCX)

**S5 Table. Supply of different food groups Bangladesh between 2010–2019 (kg/capita/year).** Source: FAOstat website: http://www.fao.org/faostat/en/#data/FBS.
(DOCX)

## Author Contributions

**Conceptualization:** Phuong Hong Nguyen, Samuel Scott.

**Data curation:** Phuong Hong Nguyen, Salauddin Tauseef, Long Quynh Khuong.

**Formal analysis:** Phuong Hong Nguyen, Long Quynh Khuong.

**Funding acquisition:** Purnima Menon.

**Investigation:** Phuong Hong Nguyen, Sk. Masum Billah, Purnima Menon, Samuel Scott.

**Methodology:** Phuong Hong Nguyen, Salauddin Tauseef, Long Quynh Khuong, Rajat Das Gupta, Sk. Masum Billah, Purnima Menon, Samuel Scott.

**Supervision:** Phuong Hong Nguyen, Purnima Menon.

**Visualization:** Long Quynh Khuong.

**Writing – original draft:** Phuong Hong Nguyen, Salauddin Tauseef, Rajat Das Gupta, Sk. Masum Billah, Samuel Scott.

**Writing – review & editing:** Phuong Hong Nguyen, Salauddin Tauseef, Long Quynh Khuong, Rajat Das Gupta, Sk. Masum Billah, Purnima Menon, Samuel Scott.

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
