## [Decision Letter · Decision Letter 0]

6 Jun 2022

PONE-D-22-13439Underweight, overweight/obesity and non-communicable diseases in Bangladesh, 2004 to 2018PLOS ONE

Dear Dr. Nguyen,

Thank you for submitting your manuscript to PLOS ONE. After careful consideration, we feel that it has merit but does not fully meet PLOS ONE’s publication criteria as it currently stands. Therefore, we invite you to submit a revised version of the manuscript that addresses the points raised during the review process.

We look forward to receiving your revised manuscript.

Kind regards,

Negar Rezaei, M.D., Ph.D.,

Academic Editor

PLOS ONE

Journal Requirements:

"Funding for this research was provided by the Bill & Melinda Gates Foundation through Alive & Thrive, managed by FHI Solutions (grant numbers: OPP1170427) and the CGIAR Research Program on Agriculture for Nutrition and Health (A4NH), led by the International Food Policy Research Institute (IFPRI)."

"Funding for this research was provided by the Gates Foundation through Alive & Thrive, managed by FHI Solutions (grant numbers: OPP1170427). The funders had no role in study design, data collection and analysis, data interpretation, decision to publish, or preparation of the manuscript."

5. We note that Figures 3 and 4 in your submission contain [map/satellite] images which may be copyrighted. All PLOS content is published under the Creative Commons Attribution License (CC BY 4.0), which means that the manuscript, images, and Supporting Information files will be freely available online, and any third party is permitted to access, download, copy, distribute, and use these materials in any way, even commercially, with proper attribution. For these reasons, we cannot publish previously copyrighted maps or satellite images created using proprietary data, such as Google software (Google Maps, Street View, and Earth). For more information, see our copyright guidelines: http://journals.plos.org/plosone/s/licenses-and-copyright.

a. You may seek permission from the original copyright holder of Figures 3 and 4 to publish the content specifically under the CC BY 4.0 license.  

Reviewers' comments:

Reviewer's Responses to Questions

**Comments to the Author**

1. Is the manuscript technically sound, and do the data support the conclusions?

Reviewer #1: Yes

Reviewer #2: Yes

Reviewer #3: Yes

Reviewer #4: Yes

2. Has the statistical analysis been performed appropriately and rigorously? 

Reviewer #1: Yes

Reviewer #2: Yes

Reviewer #3: Yes

Reviewer #4: Yes

3. Have the authors made all data underlying the findings in their manuscript fully available?

Reviewer #1: Yes

Reviewer #2: Yes

Reviewer #3: Yes

Reviewer #4: Yes

4. Is the manuscript presented in an intelligible fashion and written in standard English?

Reviewer #1: Yes

Reviewer #2: Yes

Reviewer #3: Yes

Reviewer #4: Yes

5. Review Comments to the Author

Reviewer #1: The authors of this study investigated the epidemiology of some of the key public health issues in Bangladesh, a country with major public health issues and challenges. Although the communicable and infectious diseases have been the main health issue in this country in the past decades, the urbanizing and aging population of country have led to increase in prevalence and incidence of non-communicable and chronic diseases and conditions in the past years. This study benefits a rigor data and analysis; however, the manuscript could benefit from some improvements. My comments and suggestions are provided below in this regard to help making any decision on this submission.

1. Title: the body of manuscript shows that this study evaluates a couple of conditions; however, the title is not inclusive of all these conditions and somehow is misleading. For example, using the term “non-communicable diseases” when only diabetes and hypertension is investigated in study, is confusing and inappropriate. It is suggested that authors kindly revise the title of manuscript.

2. General: language and grammar edit are necessary for the manuscript to enhance its readability.

3. Abstract: the provided results in abstract are too much making following the results hard. It is suggested that authors provide only main results here and keep the rest for the main text of manuscript.

4. Introduction: adding some information on the recent statistics of included conditions using updated references like the recent Global Burden of Diseases (GBD) 2019 estimations, for Bangladesh, could provide an interesting context in this section.

5. Methods, Determinants of malnutrition and NCDs, lines 135-147: the first part of this section seems to be irrelevant to what should be provided in this section. The provided material could be moved to introduction or discussion sections.

6. Methods: the details of the regression decomposition analysis is missing and a comprehensive expansion on this method is necessary in the corresponding part of the methods section.

7. Results, tables, and figures: well prepared.

8. Discussion: alongside the other sections of the manuscript, although this investigation explores numerous conditions and related factors and variables, the one major point which needs to be focused through the text and specifically in the discussion section is missing. The authors may choose one as the major issue and finding of this study and expand the discussion a little more ton the chosen challenge.

Reviewer #2: This study aims to explore BMI, Diabetes, and hypertension status in Bangladesh. The dataset is carefully selected for this purpose, and the chosen aims are real public health concerns. For this purpose, statistical methods and data presentation techniques (Figures) were chosen appropriately. The manuscript was written adequately in terms of introduction and methods. However, the results and discussion were not satisfactory. The reason could be establishing too many aims for a single study. Therefore, I prefer to share my critical concerns:

The aims of the study in lines 81:84 are relatively generally stated. In such research, we need to specify objectives. This issue caused other drawbacks to working.

The title does not reflect the work accurately. It contains underweight, overweight/obesity as specific terms but summarizes diabetes and hypertension in con communicable disease terms. In addition, Using "or" state (/) in the title is not attractive.

The abbreviation should be mentioned in the first place of the text. For instance, look at lines 28 and 174. There are two different approaches for SII and CIX. In addition, the common abbreviation for concentration index is (CI). In some work, the authors prefer CIX to differentiate between this and the confidence interval. It does not satisfy your assignment.

The wealth index changes a lot over the years, as you mentioned in lines 56 to 64. How could you differentiate the decrease in wealth index variation and inequality of outcomes related to WI?

You should select one of these approaches: 1- explore the trend of outcomes or 2- explore health inequality across time. Following both of these aims leads to non sufficiently discussion and conclusion for both. Depending on your choice, you need to change the initial results with supplementary results or vice versa.

It seems more informative to explore overweight and obesity as the different outcomes.

The BMI category could be a relevant factor in the decomposition of diabetes and hypertension if you focus only on these outcomes.

Table 1: Please add percentages or standard deviation

Reviewer #3: This prevalence study by Nguyen, P. H., et al., 2022 used representative data from five rounds of Bangladesh Demographic and Health Surveys (BDHS) (2004, 2007, 2011, 2014 and 2018) at both national and division levels. The survey used a two-stage stratified sampling strategy, and a sampling frame was obtained from the list of enumeration areas of the Population and Housing Census of the People’s Republic of Bangladesh, provided by the Bangladesh Bureau of Statistics.

.

Dependent (outcome) variables: underweight, overweight/obesity, hypertension, diabetes

Independent variables: categorized (in both women and men) at household level (size, SES, exposure to media, rural/urban, hygiene and sanitation) and individual level (age, education, employment).

A total of 74,798 households were surveyed across the five rounds, among them 76,667 non-pregnant women aged 15-49y and 10,900 men aged 18-95y having height and weight measurement data. The number of non-pregnant women and men aged 35-95y having data on NCDs were 7,333 and 6,821 respectively. All analyses were conducted using Stata 17 (StataCorp 187 LLC, College Station, TX, USA). Statistical significance was considered at p < 0.05. This study was a secondary data analysis of BDHS data, which was approved by the

institutional review board at ICF and the Bangladesh Medical Research Council. All respondents in the BDHS undergo an informed consent process for participation in the survey.

Results revealed that on average across survey rounds, women were 30 years old, men were 42-45 years old and most (72-78%) lived in rural areas. Over time, household size reduced

197 from 5.6 to 5.1 members and wealth index increased from 2.1 to 6.1. Hygiene and sanitation also improved by 22 percentage points (pp). Average years of education increased for women (from 3.4 years in 2004 to 5.6 years in 2018) and men (4.6 years in 2011 to 5.6 years in 200 2018). Women participating in labor workforce also increased over time.

The authors made all data underlying the findings fully available. The data was tested for representativeness, analyzed using descriptive and inferential statistics which were rigorous and appropriate.

Discussions of the results were robust, citing similar studies conducted both within and outside Bangladesh.

Conclusions are in line with the findings

Writing quality and clarity: Satisfactory

Other observations:

1. Limitations of the study: The authors did well to mention the limitations of the study

2. Inclusion/exclusion criteria not clearly explained as a separate sub topic

References: The manuscript employed the use of Harvard style referencing but requires editing to correct some errors noticed e.g., Listing of references: Shouldn’t this be in alphabetical order? Shouldn’t the journal name be italics? Shouldn’t the list of authors that are more than 5 be reflected as et al?

I suggest the authors should revise Harvard referencing style and make necessary corrections.

Reviewer #4: The authors assessed the trends of non-communicable diseases and factors that have contributed to NCDs in Bangladesh. This study could have a substantial impact on improving the health system in Bangladesh.

I would suggest some points below:

General comment:

Tables and Figures need a detailed description in the text or a detailed caption. For example, Table 1 needs an explanation of the summary statistic based on variables type.

Methods:

1) page 5, lines 98-99: an explanation of the reason for choosing a subsample of men only, and not including women.

2) The details of variables can be transferred to the supplementary appendix.  

3) To address the potential bias related to each data set in the analysis, you need to undertake a sensitivity analysis.

4) In table 2, the number and percentage of individuals with the outcome need to be added as separate columns for each NCD's risk factor.

6. PLOS authors have the option to publish the peer review history of their article (what does this mean?). If published, this will include your full peer review and any attached files.

Reviewer #1: **Yes: **Sina Azadnajafabad, MD, MPH

Reviewer #2: No

Reviewer #3: **Yes: **Haruna Ismaila ADAMU: MD; MPH; PhD; MACE

Reviewer #4: No

---

## [Author Response · Author response to Decision Letter 0]

2 Aug 2022

Please see the Responses to Reviewers attached separately.

---

## [Decision Letter · Decision Letter 1]

4 Sep 2022

PONE-D-22-13439R1Underweight, overweight or obesity, diabetes and hypertension in Bangladesh, 2004 to 2018PLOS ONE

Dear Dr. Nguyen,

Thank you for submitting your manuscript to PLOS ONE. After careful consideration, we feel that it has merit but does not fully meet PLOS ONE’s publication criteria as it currently stands. Therefore, we invite you to submit a revised version of the manuscript that addresses the points raised during the review process.

We look forward to receiving your revised manuscript.

Kind regards,

Negar Rezaei, M.D., Ph.D.,

Academic Editor

PLOS ONE

Journal Requirements:

Reviewers' comments:

Reviewer's Responses to Questions

**Comments to the Author**

1. If the authors have adequately addressed your comments raised in a previous round of review and you feel that this manuscript is now acceptable for publication, you may indicate that here to bypass the “Comments to the Author” section, enter your conflict of interest statement in the “Confidential to Editor” section, and submit your "Accept" recommendation.

Reviewer #1: All comments have been addressed

Reviewer #2: All comments have been addressed

Reviewer #3: All comments have been addressed

Reviewer #4: All comments have been addressed

2. Is the manuscript technically sound, and do the data support the conclusions?

Reviewer #1: Yes

Reviewer #2: Partly

Reviewer #3: Yes

Reviewer #4: Yes

3. Has the statistical analysis been performed appropriately and rigorously? 

Reviewer #1: Yes

Reviewer #2: Yes

Reviewer #3: Yes

Reviewer #4: Yes

4. Have the authors made all data underlying the findings in their manuscript fully available?

Reviewer #1: Yes

Reviewer #2: Yes

Reviewer #3: Yes

Reviewer #4: Yes

5. Is the manuscript presented in an intelligible fashion and written in standard English?

Reviewer #1: Yes

Reviewer #2: No

Reviewer #3: Yes

Reviewer #4: Yes

6. Review Comments to the Author

Reviewer #1: The authors addressed the comments adequately and provided reasonable responses in this revision and the response letter, regarding my comments. The manuscript is acceptable in this format in my opinion.

Reviewer #2: The authors answered my concerns carefully. I want to thank them. However, there are several concerns:

It is arbitrary in Plos one style to write the abstract in a structured format. However, adding a title for each part make the abstract more readable.

The age period cohort effect is demonstrated in figure 1. It would help if you discussed this effect more.

The higher degree of smoothness in statistical models could be misleading. Adding observed points to the smooth lines in Figures 1 and 2 could be informative.

Reviewer #3: All my previous comments have been adequately addressed. The paper is now Ok and acceptable for publication

Reviewer #4: The authors perfectly addressed the comments, which improved the results and methodology.

I would suggest including the percentage [number(and %)] of individuals at each study round contributed to the whole analysis. For example, the number of individuals in 2004 and the percentage of the data for this year compared to the total number of individuals included in the analysis. It helps to have an overview of the study size in one look at the table.

7. PLOS authors have the option to publish the peer review history of their article (what does this mean?). If published, this will include your full peer review and any attached files.

Reviewer #1: **Yes: **Sina Azadnajafabad, MD, MPH

Reviewer #2: No

Reviewer #3: **Yes: **Haruna Ismaila ADAMU, MBBS; MPH; PhD

Reviewer #4: No

---

## [Author Response · Author response to Decision Letter 1]

10 Sep 2022

Please see the Responses to Reviewers attached separately

---

## [Editor Report · Decision Letter 2]

13 Sep 2022

Underweight, overweight or obesity, diabetes and hypertension in Bangladesh, 2004 to 2018

PONE-D-22-13439R2

Dear Dr. Nguyen,

We’re pleased to inform you that your manuscript has been judged scientifically suitable for publication and will be formally accepted for publication once it meets all outstanding technical requirements.

Kind regards,

Negar Rezaei, M.D., Ph.D.,

Academic Editor

PLOS ONE
---

## [Editor Report · Acceptance letter]

22 Sep 2022

PONE-D-22-13439R2 

Underweight, overweight or obesity, diabetes, and hypertension in Bangladesh, 2004 to 2018 

Dear Dr. Nguyen:

I'm pleased to inform you that your manuscript has been deemed suitable for publication in PLOS ONE. Congratulations! Your manuscript is now with our production department. 

Kind regards, 

on behalf of

Dr. Negar Rezaei 

Academic Editor

PLOS ONE